# GUIDED SAMPLING IN REINFORCEMENT LEARNING FOR LLM REASONING

## ABSTRACT

Reinforcement Learning (RL) has proven effective at fine-tuning Large Language Models (LLMs) to improve the precision of their chain-of-thought reasoning. However, these methods typically rely on outcome-based rewards, without directly supervising the cognitive process of reflection. Consequently, while the model's ability to complete reasoning tasks is optimized, its capacity to identify and recover from its own errors within a single, continuous line of thought is not explicitly trained. In this work, we introduce Guided Sampling, a framework designed specifically to cultivate this missing reflection ability. Guided sampling casts the exploration phase as a sequential process where, upon generating an incorrect response, the model is prompted to re-evaluate its flawed reasoning and continue its generation. This technique creates a direct optimization pressure on the act of reflection itself, shifting the learning objective from merely finding a correct answer to actively correcting a wrong one. Experimental results demonstrate that by explicitly training for reflection, our GSRL framework is able to not only surpass traditional RL methods in final task accuracy but also fosters a more robust, self-correcting reasoning process.

## 1 INTRODUCTION

Large Language Models have achieved strong performance across diverse NLP tasks, yet reliably executing complex, multi-step reasoning in mathematics, logic, and programming remains challenging. Recent work increasingly turns to reinforcement learning (RL) for post-training, where policy-gradient methods such as PPO Schulman et al. (2017) and critic-free GRPO Shao et al. (2024) are used to optimize verifiable objectives (e.g., correctness of final answers) and to encourage structured *chain-of-thought* (CoT) reasoning Wei et al. (2022); Guo et al. (2025). In practice, verifiers and programmatic reward checkers make outcome supervision feasible for math and code Hendrycks et al.

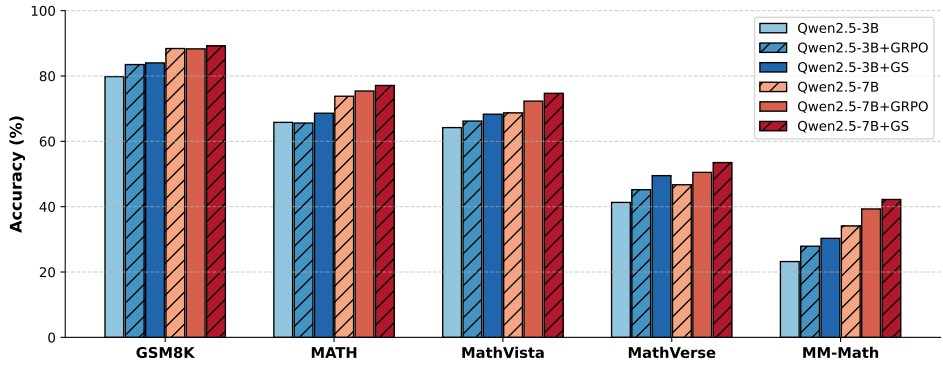

Figure 1: Guided Sampling improves both text and multi-modal reasoning across benchmarks. Compared to vanilla GRPO, GS yields consistent gains on GSM8K, MATH, MathVista, MathVerse, and MM-Math for both 3B and 7B Qwen2.5-VL backbones, highlighting the benefit of guided LLM reflection.

(2021); Cobbe et al. (2021); Jain et al. (2024), while scalable systems enable on-policy training at scale Yu et al. (2025); Ahmadian et al. (2024); Hu (2025).

A standard RL pipeline for reasoning samples multiple responses per query and then rewards each trajectory according to the correctness of its final outcome. While effective for improving pass@1 and pass@k Chen et al. (2025), this paradigm faces two central limitations.

First, RL like GRPO focuses optimization on the outcome correctness. In practice, correct answers are often concise while incorrect attempts are longer; thus, GRPO tends to favor short answers that offer little incentive to develop robust chain-of-thought or self-reflection behaviors. To compensate, prior works Shinn et al. (2023); Wan et al. (2025); Madaan et al. (2023); Ji et al. (2023) inject reflection via prompting or SFT to enforce a "answer-then-reflect" format of LLM. However, rigid, template-driven reflections introduce several drawbacks: (i) they can induce inefficient thinking and even degrade the reasoning ability by overthinking. (ii) there are chances that the previous correct answer is overwritten by the reflection process and injects noisy supervision to RL training. Moreover, recent analyses further show that outcome-only RL can collapse output diversity, degrading pass@k at larger k and hurting test-time scaling Chen et al. (2025); Song et al. (2025). These issues jointly motivate a training paradigm that preserves outcome-grounded supervision and improves the self-reflection ability.

Second, existing reflection-aware RL typically scores entire trajectories with sparse outcome signals, sometimes adding a bonus when a reflection turns an incorrect answer into a correct one Shinn et al. (2023); Wan et al. (2025); Kumar et al. (2024). However, this trajectory-level scoring leads to weak credit assignment for the specific thinking and reflection that actually drive improvement. In practice, reflections not always improves over the intial response, collapsing the think and reflection into a single outcome reward obscures the contribution of each parts. Moreover, sparse terminal rewards exacerbate variance under on-policy sampling Ahmadian et al. (2024), and outcome-only shaping encourages premature truncation of reasoning traces Guo et al. (2025); Liu et al. (2025).

We address these limitations by reframing exploration and learning as a sequential, reflective process. We introduce *Guided Sampling*, an RL sampling pipeline that casts reasoning as guided, multi-step decisions over *attempts* and *reflections*. During policy optimization, rather than drawing $N$ independent responses from a fixed model policy, Guided Sampling keeps each failed attempt as the context and explicitly prompts the model to review, diagnose, and repair its reasoning. This turns sparse, terminal outcomes into dense, mid-trajectory signals that provide clearer credit assignment to both thinking and reflection. By optimizing the reflection policy alongside outcome correctness, Guided Sampling. See Figure 1 for a summary of accuracy gains across GSM8K, MATH, MathVista, MathVerse, and MM-Math on Qwen2.5-VL 3B/7B backbones, where Guided Sampling consistently outperforms vanilla GRPO.

Our contributions are:

1. We introduce **Guided Sampling**, a novel RL sampling pipeline that augments models exploration by explicitly triggering reflection for failed attempts for model reflection training.

2. We design segment-level rewards that disentangle credit for initial reasoning from credit for subsequent reflections, improving credit assignment and policy optimization for both behaviors.

3. Experimental results across text-only and multi-modal reasoning benchmarks show that Guided Sampling surpasses vanilla GRPO in final accuracy and, critically, more robust self-correcting reasoning.

## 2 PRELIMINARY

Group Relative Policy Optimization (GRPO) Shao et al. (2024) is a reinforcement learning algorithm designed to optimize language models efficiently without requiring an explicit value function or critic. GRPO modifies the standard Proximal Policy Optimization (PPO) objective by computing token-level advantages based on group-normalized rewards, making it particularly suitable for LLMs.

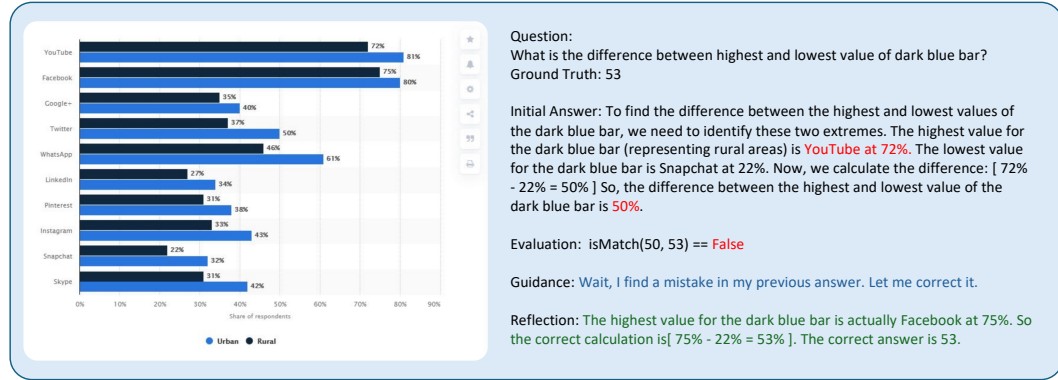

Figure 2: An example of Guided Sampling at GRPO training time. Given a question, the model first generates an initial solution attempt. If incorrect, a lightweight guidance cue is appended to the failed attempt to trigger a reflection behavior. Unlike Best-of-N, Guided Sampling is able to explicitly create self-correction data for reflection training. The illustrated example shows the model identifying a mistaken extreme in a chart question, triggering reflection, and revising the computation to the correct result.

Given a batch of $G$ responses $\{o_i\}_{i=1}^G$ from a query $q$, each consisting of a sequence of tokens $o_i = (o_i(1), ..., o_i(T))$, the GRPO objective is defined as:

$$J(\theta) = \frac{1}{G} \sum_{i=1}^G \frac{1}{|o_i|} \sum_{t=1}^{|o_i|} \min \left[ \frac{\pi_\theta(o_i(t)|o_{i,<t})}{\pi_{\text{old}}(o_i(t)|o_{i,<t})} \hat{A}_{i,t}, \text{clip} \left( \frac{\pi_\theta(o_i(t)|o_{i,<t})}{\pi_{\text{old}}(o_i(t)|o_{i,<t})}, 1 - \varepsilon, 1 + \varepsilon \right) \hat{A}_{i,t} \right], \tag{1}$$

where $\hat{A}_{i,t}$ is the group-normalized advantage for token $t$ in response $o_i$, computed as:

$$\hat{A}_{i,t} = \frac{r_i - \mu}{\sigma}, \quad \text{with } r_i \text{ the total reward of } o_i, \tag{2}$$

and $\mu$, $\sigma$, the mean and standard deviation of rewards in the group.

## 3 METHOD

In this section, we present the detailed methodology of our Guided Sampling framework. We describe the reflection trigger used to shift the model to reflection thinking. Next, we detail the reward decomposition and advantage routing specifically for our training data pipeline. Finally, we present the training protocols and implementation specifics for GRPO optimization.

### 3.1 GUIDED SAMPLING: A SEQUENTIAL, REFLECTIVE PROCESS

Guided Sampling reframes the exploration phase from a set of parallel, independent trials into a single, sequential Markov Decision Process (MDP). Instead of discarding a failed attempt, we use it as an intermediate state and explicitly prompt the model to reflect and continue.

The core of our method is the **Reflection Trigger**, a simple mechanism activated when a generated response is evaluated as incorrect, see Figure 2. Instead of terminating the generation, we explicitly append a reflection prompt to the failed trajectory and ask the model to continue its generation.

To achieve this, we curate a list of naturalistic reflection prompts, such as:

- "Now let me review the result. That doesn't seem right."
- "Wait, let me re-evaluate my reasoning from the start."
- "Let me double-check my work. I think there might be a mistake in the previous steps."

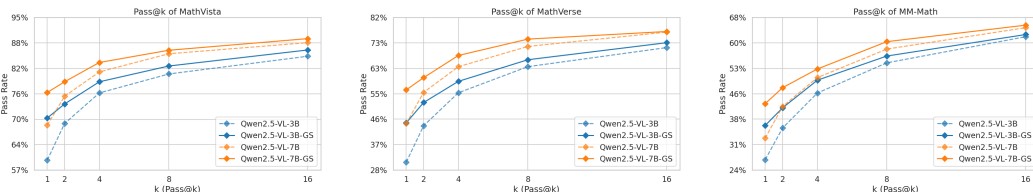

Figure 3: Comparison of Pass@K between BoN and Guided Sampling at training time. Guided Sampling consistently outperforms BoN when the evaluation of the sample is available during sampling.

Motivated by the effectiveness of conversational cues in models like DeepSeek Guo et al. (2025), we heuristically inject words like "Wait," into the prompts to effectively shift the model from a generative state to a more critical, evaluative one. When a response $y_{\text{fail}}$ is generated and deemed incorrect, the new input to the model becomes $\texttt{concat}(q, y_{\text{fail}}, g_{\text{reflection}})$. This transforms a failure from an endpoint into a learning opportunity, creating a direct optimization pressure on the act of self-correction. We focus on enabling the model to learn this reflective behavior in the natural language space, rather than forcing it into a rigid format via Supervised Fine-Tuning.

### 3.2 THE MULTI-TURN ANSWER REFINEMENT LOOP

We model the refinement process as a sequential decision-making problem. For a given query $q$, the model generates a sequence of actions and receives corresponding rewards: $(a_1, r_1, a_2, r_2, \ldots, a_T, r_T)$, where $T$ is the maximum number of turns.

**Multi-turn Reflection.** During RL training, we implement Guided Sampling as a multi-turn refinement loop. The process for each query proceeds as follows: The initial state is the query, $s_0 = q$. At the first turn, the model takes an action $a_1$ by generating an initial response, $y_1 \sim \pi_\theta(\cdot|s_0)$. This action is immediately evaluated, yielding a reward $r_1 = R(y_1)$. If the response is correct ($r_1 = 1$), the episode terminates successfully. If the response is incorrect ($r_1 = 0$), the episode continues. The state is updated by appending the failed response and a reflection prompt, creating a new state $s_1 = \texttt{concat}(s_0, y_1, g_{\text{reflection}})$. The model then takes a second action $a_2$ by generating a new response $y_2 \sim \pi_\theta(\cdot|s_1)$. The action $a_2$ represents the entire act of generating a refined answer based on the history of failure and the explicit prompt to self-correct. This process of action, evaluation, and state transition repeats for a maximum of $T$ turns. The full trajectory, containing the history of attempts and reflections, is then used to update the policy $\pi_\theta$. This framework uses the intermediate rewards $(r_1, r_2, \ldots)$ as signals to trigger reflection, directly training the model to recover from errors within a single, continuous line of reasoning.

**Relation to Best-of-N.** Our training methodology is fundamentally different from Best-of-N (BoN) sampling, and we hypothesize it leads to a more capable model. BoN is an *inference-time* strategy that generates $N$ independent samples and selects the best one, relying on brute-force exploration. In contrast, Guided Sampling is a *training-time* strategy that teaches the model how to intelligently navigate the solution space. For a comparable budget of $N$ attempts, BoN generates $N$ full, parallel trajectories, hoping one is correct. Guided sampling generates trajectories in a sequential way under the guidance of the intermediate reward signal.

The key distinction lies in the learning signal: in Guided Sampling, the intermediate rewards ($r_t < 0.5$) actively trigger a state change and prompt the model to correct a *specific, observed error*. This creates a rich, contextual learning signal for self-correction that is absent in BoN sampling, where failed trajectories are simply discarded after providing a low-reward signal. Since the ground truth is used for intermediate evaluation, Guided Sampling inherently outperforms BoN sampling. Figure 3 show empirically that Guided Sampling under the same rollout numbers.

### 3.3 Training Details for Guided Sampling

**Reward decomposition and normalization.**  For each query, a verifier assigns rewards $r_{init}$ and $r_{ref}$ separately to the initial attempt $o_{init}$ and to each reflection answer $o_{ref}$. We then normalize rewards jointly within the GRPO group according to Equation 2. All rewards include first pass thinking and reflection are pooled to compute a single mean and standard deviation, and every token uses this shared normalization. This establishes a common scale across tokens while we still route credit at the token level using segment-aware advantages, so thinking tokens do not take credit for reflection fixes and vice versa.

**Advantage assignment**  In Guided Sampling, each query can include additional guidance and reflection token segments during GRPO training. Rather than assigning the same advantage to all tokens as in vanilla GRPO, we decompose the rewards and route credit segment-wise: a token only receives advantage from the reward that follows its segment. This yields a cleaner learning signal than applying a weighted sum of multiple rewards to the entire sequence. For a sequence of tokens that consists of a normal first-pass answer $o_{init}$ and a reflection $o_{ref}$, their optimization target is separately computed using Equation 1.

**Disable clipping for reflection guidance.**  Naively concatenating heuristically designed reflection prompts can increase PPO clipping on tokens that are distributionally shifted relative to the base policy. Thus the predicted log-probability will be small. Similar to the principle of DAPO Yu et al. (2025), we relax clipping on guidances and reflection tokens by increasing the upper clip bound $\epsilon_{high}$ while keeping standard clipping for regular tokens. This prevents premature truncation caused by predicting the end-of-text token other than guidance tokens at test time, which help sustain exploration in the reflective mode, and mitigates rapid entropy collapse, leading to more stable and effective reflection learning.

## 4 Experiments

### 4.1 Implementations

**Training Datasets and Benchmarks.**  For multi-modal RL training, we adopt 10k multi-modal math questions from the training set of Vision-R1 Huang et al. (2025). We evaluate the baselines and our model on MathVista Lu et al. (2023), MathVerse Zhang et al. (2024), and MM-Math Sun et al. (2024). For text-only RL training, we adopt the training split of GSM8K Cobbe et al. (2021) and Math Hendrycks et al. (2021) and evaluate on their test set.

**Implementation Details.**  We take Qwen2.5-Instruct Team (2024b) and Qwen2.5-VL-Instruct Bai et al. (2025) as our baseline models and finetune with VERL framework Sheng et al. (2025). We train the RL model with 30 epochs for multi-modal reasoning. We set the training batch size to 2048 and micro batch size per gpu to 64 and 4 for text-only task and vision task. We set the rollout number per query to 8. The learning rate is set to be $1 \times 10^{-6}$. We use Adam Optimizer Kingma & Ba (2014) for training. During training, we allow each question to reflect at most once. Additional hyper-parameter settings and detailed prompt configurations are provided in Appendix.

### 4.2 Multi-Modal Reasoning Results

Table 1 shows that Guided Sampling (Ours-7B) delivers consistent gains over GRPO-7B across all benchmarks and MathVista sub-domains. On MathVista-ALL, MathVerse, and MM-Math, our method improves absolute accuracy by +2.4, +3.0, and +2.9 points, respectively, raising the overall average from 51.9 to 56.7. The gains are uniform across MathVista categories, GEO (+5.5), ARI (+4.9), GPS (+4.7), and MWP (+4.2), indicating that converting failed attempts into reflective continuations trains broadly applicable recovery behaviors rather than overfitting to any single format. Against strong 7B baselines, Ours-7B surpasses Vision-R1-7B on MathVista (74.7 vs. 73.5), Math-Verse (53.5 vs. 52.4), and MM-Math (42.2 vs. 40.2), achieving a +2.9 average improvement over GRPO.

Despite using a 7B backbone, Guided Sampling narrows the gap to much larger models. It even outperforms Qwen2.5-VL-72B on MathVerse (53.5 vs. 51.3) while remaining competitive on MathVista-ALL. The largest absolute gains appear in geometry and arithmetic, tasks prone to local errors where reflection-triggered repair is most beneficial. Overall, the results support our claim that modeling exploration as a sequential, reflective process yields robust within-trajectory self-correction and higher accuracy under comparable compute budgets.

Table 1: Model Performance Comparison on Various Multi-modal Math Reasoning Benchmarks

| Model | MathVista | | | | | MathVerse | MM-Math | AVG. |
|---|---|---|---|---|---|---|---|---|
| | GEO | ARI | GPS | MWP | ALL | | | |
| Hint-GRPO | – | – | – | – | 54.2 | 32.2 | 8.7 | 31.7 |
| R1-VL-7B | – | – | – | – | 63.5 | 40.0 | 24.5 | 42.6 |
| R1-Onevision-7B | – | – | – | – | 64.1 | 46.4 | 29.2 | 46.6 |
| Qwen2.5-VL-7B | 66.9 | 68.7 | 66.8 | 76.9 | 68.1 | 46.7 | 34.1 | 47.9 |
| Qwen2.5-VL-72B | 77.8 | 77.9 | 78.8 | 74.7 | 73.5 | 51.3 | 45.6 | 55.8 |
| Vision-R1-7B | 80.3 | 79.0 | **83.2** | 80.6 | 73.5 | 52.4 | 40.2 | 53.8 |
| GRPO-7B | 76.6 | 75.4 | 78.3 | 78.0 | 72.3 | 50.5 | 39.3 | 51.9 |
| Ours-7B | **82.1** | **80.3** | 83.0 | **82.2** | **74.7** | **53.5** | **42.2** | **56.7** |

## 4.3 Text-only Reasoning Result

As presented in Tab. 2, Guided Sampling consistently outperforms both the base instruct models and GRPO across GSM8K and MATH-500 benchmarks. On the smaller 3B backbone, it raises GSM8K from 83.5 to 84.0 and MATH-500 Avg from 65.6 to 68.6, especially with larger gains on difficult levels such as L3 from 80.0 to 83.8 and L5 from 32.1 to 41.0. On the larger 7B backbone, it increases GSM8K from 88.3 to 89.2 and MATH-500 Avg from 75.4 to 77.1, while pushing L5 from 50.0 to 51.5. These results show that sequential, reflection-driven exploration strengthens single-try mathematical reasoning, especially on the difficult tiers.

## 4.4 Ablation Study

**Ablation** Table 3 reports gains on MATH-500 with Qwen2.5-3B. GRPO alone slightly drops from 65.8 to 65.6, demonstrating that outcome-only RL does not necessarily improve the chain of thought result. Adding the guidance mechanism raises the pass rate to 67.4 by turning failures into mid-trajectory signals. Clip-rate stabilization adds a small but consistent boost to 67.7 by preventing degeneration on reflection segments. Segment-level advantages give the final lift to 68.6 through precise credit assignment to initial attempts and to reflections.

**Reflection Success Rate.** To fairly assess reflection optimization, we construct an evaluation set of 311 samples from MathVista consisting only of first-turn failures produced by Qwen2.5-3B. We compare three test-time settings: (i) Force-GS, which applies the guidance prompt to the untrained Qwen2.5-3B and forces a reflection generation from the failed attempt; (ii) Direct

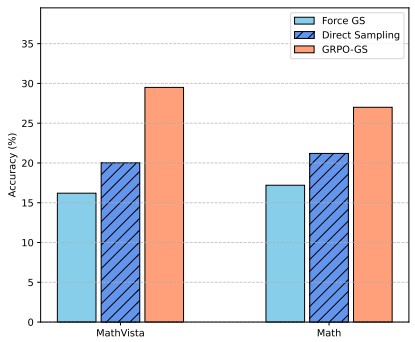

Figure 4: Reflection success rates on MathVista and MATH-500.

Sampling, which asks the model to answer again from scratch; and (iii) GRPO-GS, which performs reflection with our model trained under Guided Sampling. We rollout and collect 16 responses for each setting.

Figure 4 shows that forcing the base model to reflect hurts accuracy relative to simply re-answering. On MathVista, Force-GS attains 16 percent, Direct Sampling 20 percent, while GRPO-GS reaches

Table 2: Text-only reasoning results on GSM8K and MATH-500. We report Average Accuracy for GSM8K and Average Pass@1 for MATH-500, along with difficulty-wise breakdown on MATH-500.

| Model | GSM8K | MATH-500 by Level | | | | | |
|---|---|---|---|---|---|---|---|
| | | Avg | L1 | L2 | L3 | L4 | L5 |
| Qwen2.5-3B-Instruct | 79.8 | 65.8 | 86.0 | 85.6 | 80.0 | 60.2 | 40.3 |
| + GRPO | 83.5 | 65.6 | **97.7** | 86.7 | 80.0 | **63.3** | 32.1 |
| + Guided Sampling | **84.0** | **68.6** | 95.5 | **87.8** | 83.8 | 62.5 | **41.0** |
| Qwen2.5-7B-Instruct | 84.2 | 73.8 | 97.7 | 90.0 | 86.7 | 70.3 | 48.5 |
| + GRPO | 88.3 | 75.4 | **95.4** | 93.3 | 87.6 | 72.7 | 50.0 |
| + Guided Sampling | **89.2** | **77.1** | 93.0 | **93.3** | 89.5 | 74.2 | 51.5 |

| | GRPO | Guidance | Clip | Adv. | MATH-500 |
|---|---|---|---|---|---|
| Qwen2.5-3B | | | | | 65.8 |
| + GRPO | ✓ | | | | 65.6 |
| + Guidance | ✓ | ✓ | | | 67.4 |
| + Clip | ✓ | ✓ | ✓ | | 67.7 |
| + Adv. | ✓ | ✓ | ✓ | ✓ | **68.6** |

Table 3: Ablation results for the Guidance Sampling components.

29.5 percent, a gain of roughly 13.5 points over the forced-reflection baseline and 9.5 points over re-answering. On MATH-500, Force-GS yields 17 percent, Direct Sampling 21.2 percent, and GRPO-GS 27.1 percent, improving by about 10.1 and 5.9 points respectively. These results indicate that reflection must be trained rather than merely prompted: Guided Sampling converts reflection into a reliable corrective behavior, substantially increasing the probability of recovering from an initial error.

## 4.5 TRAINING STATISTICS

**Training Statistics**    We report four diagnostics over training: (i) training reward, see Fig. 5, (ii) KL loss to the reference policy, see Fig. 6, (iii) per-step clip magnitude, see Fig. 7, and (iv) validation accuracy on MATH, see Fig. 4.5. Guided Sampling achieves consistently higher training rewards than GRPO across steps, indicating faster and stronger optimization. During training, the KL Loss is higher than GRPO due to the introduction of guidance and reflection tokens. Figure 7: Clip magnitudes are larger and more variable under Guided Sampling, consistent with relaxed clipping on reflection segments. On MATH validation, Guided Sampling steadily outperforms GRPO at all checkpoints, yielding a higher final accuracy.

**Response Length**    In Fig. 9, we show that in both text-only and vision settings, Guided Sampling consistently maintains longer responses throughout training compared to vanilla GRPO, indicating that our reflection-triggered continuations counteract the well-documented tendency of outcome, only RL to shorten CoT traces; specifically, after an initial stabilization phase, Guided Sampling plateaus at a higher token count ($\approx$450 vs. $\approx$370 for text; $\approx$400 vs. $\approx$320 for vision), and even exhibits a late-stage uptick in length for text-only training, suggesting that the model increasingly allocates budget to diagnosis-and-repair segments rather than prematurely truncating reasoning, evidence that the segment-level rewards encourage sustained, deliberative computation aligned with in-trajectory self-correction rather than terse, outcome-chasing outputs.

## 4.6 VISUALIZATION

Figure 10 presents concise, side-by-side examples showing the reflection process of Qwen2.5-7B-VL trained with our Guided Sampling. Each panel shows an initial attempt with its key error highlighted in red. The model tries to fix its error by first generating a cue that triggers reflection (e.g., "Wait, let me re-check. . . "), and the self-reflection CoT leading to the final answer. Color cues sepa-

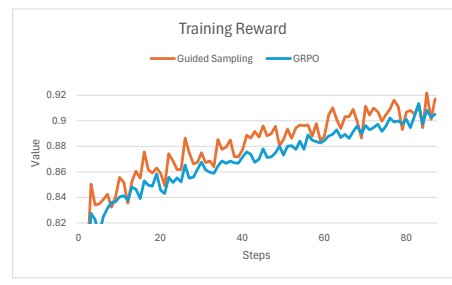

Figure 5: Training reward comparison.

Figure 6: KL Loss comparison.

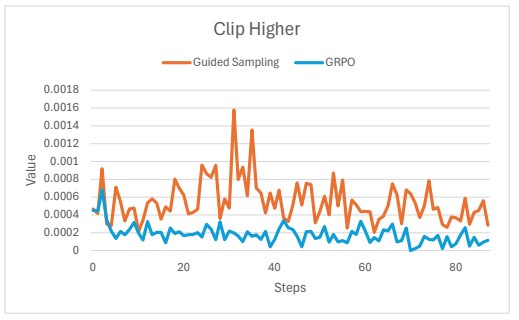

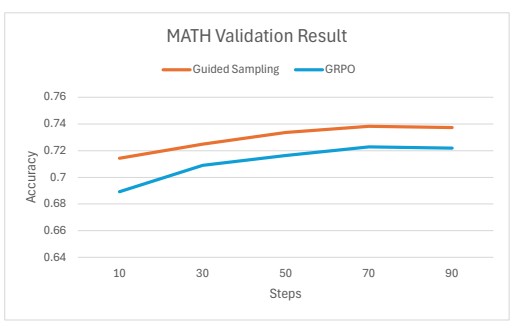

Figure 7: Clip Value comparison.

Figure 8: Performance on MATH dataset.

rate thinking and reflection tokens, and callouts mark where reward switches from failure to success. Reflections are short and task-grounded (fixing a misapplied theorem or a wrong hypotenuse), preserve correct partial derivations, and revise only faulty steps. These qualitative behaviors mirror our quantitative results, indicating that Guided Sampling turns terminal failures into focused state transitions that reliably yield concise, effective self-corrections.

## 5 RELATED WORKS

**LLM Reasoning and Reinforcement Learning.** Chain-of-thought (CoT) prompting improves multi-step problem solving but does not reliably yield within-trajectory error correction behavior Wei et al. (2022); Huang et al. (2023). Iterative self-revision and agent-style feedback Shinn et al. (2023) can recover from mistakes across attempts, yet tend to be brittle without dedicated training for the reflection behavior itself Madaan et al. (2023); Shinn et al. (2023); Ji et al. (2023). Reinforcement learning offers a principled route to optimizing verifiable reasoning with policy-gradient methods such as PPO Schulman et al. (2017) and critic-free GRPO Shao et al. (2024), where careful baseline design often yields strong outcomes Li et al. (2023); Ahmadian et al. (2024); Hu (2025). Recent studies emphasize incentivizing reasoning patterns Guo et al. (2025); Chu et al. (2025) while also surfacing stability and scaling caveats under RL Liu et al. (2025); Li et al. (2025). For math and code, programmatic verifiers enable precise outcome supervision Hendrycks et al. (2021); Lu et al. (2023); Zhang et al. (2024); Sun et al. (2024); Jain et al. (2024). Besides, there are text and multi-modal reasoning datasets and benchmarks Lu et al. (2023); Sun et al. (2024); Zhang et al. (2024); Huang et al. (2025); Wei et al. (2023); Qiao et al. (2024); He et al. (2024). Our approach differs by explicitly turning a failed attempt into a new state that triggers guided reflection within the same trajectory, producing dense, turn-level learning signals without requiring a critic.

**RL Sampling Strategies.** Best-of-$N$ (BoN) at inference improves accuracy by brute-force selection but discards failed trajectories and provides weak learning signals Sessa et al. (2024); Amini et al. (2024). Training-time alternatives focus on efficiency, competence-aware sampling, and scalable systems Kong et al. (2025); Yu et al. (2025). Reflection-aware RL introduces structured revision during optimization, including multimodal variants Wan et al. (2025); Kumar et al. (2024). Unlike parallel, independently sampled rollouts from the same distribution, our method frames exploration

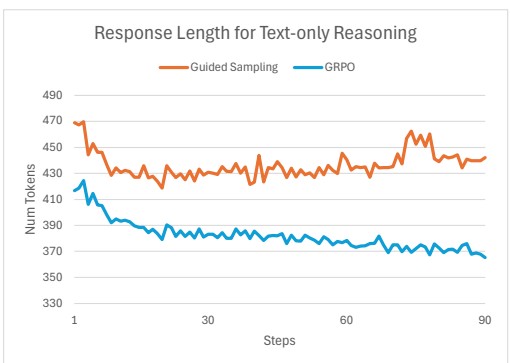
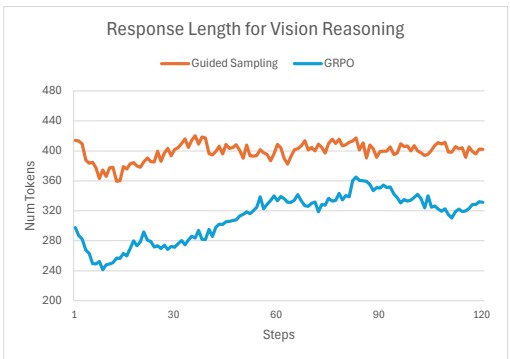

Figure 9: Comparison of response length of Qwen2.5 3B model during GRPO training for text-only reasoning (left) and vision reasoning (right).

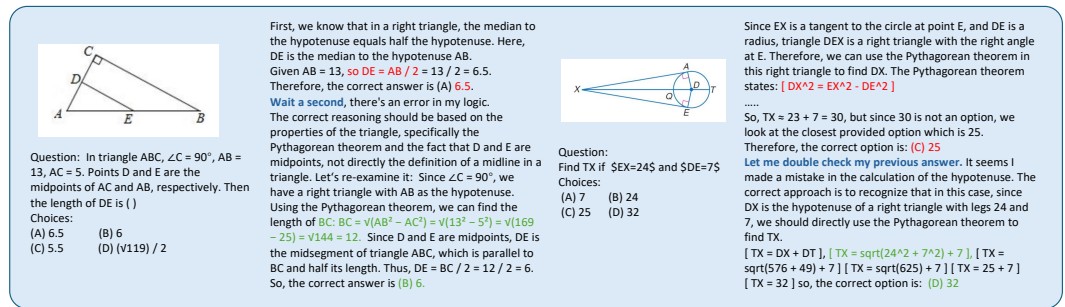

Figure 10: A step-by-step view of Guided Sampling in action: the model's initial attempt, a guidance cue that triggers reflection, and the corrected reflection to a verified answer. The key parts of mistakes and reflection are highlighted in red and green.

as a sequential process: an incorrect answer becomes the next state; a lightweight guidance cue elicits reflection; and the policy is optimized on continuation-based trajectories with credit assignment. This directly optimizes mid-trajectory self-correction and is compatible with GRPO-style updates.

# 6    CONCLUSION

In this work, we introduced Guided Sampling, a training time sampling mechanism for reflection-centered LLM reinforcement learning. By heuristically triggering reflection for failed attempts, our method supplies dense, mid-trajectory credit assignment that explicitly optimizes both thinking and reflection. Guided Sampling yields consistent gains over vanilla GRPO across text-only and multimodal reasoning benchmarks, improving both pass@1 accuracy and the reflection quality. Empirically, it sustains longer, more deliberate reasoning traces and outperforms BoN brute-force exploration under comparable sampling budgets, demonstrating that training the reflection policy itself fosters robust, self-correcting reasoning.

**Limitations.**    Our approach relies on access to reliable intermediate evaluators or verifiers to trigger reflection, which may not be available for open-ended tasks, noisy domains, or those lacking programmatic reward signals. Additionally, the method is not well-suited for environments with high-latency feedback, as the sequential, multi-turn reflection loop introduces additional computational cost and response time. As a future direction, we are interested in explicitly modeling high-level reasoning actions—such as reflection, retracing, and revision—and leveraging Monte Carlo Tree Search (MCTS) to explore the structured answer space. By integrating MCTS with reinforcement learning, we aim to train LLMs that can search, reflect, and refine more intelligently over long-horizon reasoning trajectories.

## ETHICS STATEMENT

**Scope of data and human subjects.**    We do not collect new human-subjects data and do not employ crowd annotators. All datasets used (GSM8K, MATH, MathVista, MathVerse, MM-Math) are publicly available research datasets; we followed their licenses and terms of use.

**Privacy and data governance.**    We used only public datasets within their intended research scope. Dataset licenses and usage restrictions were respected.

**Avoiding harm and dual use.**    Our method improves verifiable mathematical and multimodal reasoning via reflection. Potential misuse is mitigated by recommending deployment only with appropriate safeguards and monitoring in sensitive domains.

**Fairness and non-discrimination.**    The paper has nothing related to unfairness and discrimination.

**Environmental impact.**    We report major training details (epochs, batch sizes, rollouts, learning rates, backbones) to aid energy accounting. We reuse pretrained models and limit training epochs.

**Conflicts of interest and sponsorship.**    We will disclose any funding sources, affiliations, and potential conflicts in the camera-ready version. No sponsor influenced experimental design, analysis, or reporting.

**Research integrity.**    We report methods and results honestly, avoid fabrication or selective reporting, and credit prior work and external assets. No new human-subjects experiments were conducted; therefore, no IRB approval was required.

## REPRODUCIBILITY STATEMENT

We provide anonymous codes with complete training and evaluation code for our work, including scripts, configs, and exact prompts used. We will release trained model weights, full source code, and training logs, along with step-by-step guidance to reproduce results. The repository details datasets and splits for GSM8K, MATH, MathVista, MathVerse, and MM-Math, preprocessing, all hyperparameters such as epochs, batch sizes, rollout counts, optimizer, learning rate, KL and clip settings. We include evaluation scripts to regenerate tables and figures and instructions for environment setup to ensure repeatability.

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

## A  LLM Usage

We used LLM solely as a general-purpose writing assistant to polish prose. Specifically, after drafting all technical content (methods, experiments, figures, tables, and references) ourselves, we employed the LLM to suggest alternative phrasings, improve clarity and flow, and correct minor grammatical issues. The LLM did not contribute to research ideation, problem formulation, experimental design, implementation, data analysis, or result interpretation. All technical claims, equations, and citations were authored and verified by the authors. The authors take full responsibility for the contents of this paper. LLMs were not used to generate novel scientific content, and they are not eligible for authorship.

## B  Implementation Details

All experiments are conducted using the VERL framework, with the Qwen2.5 and Qwen2.5-VL as the base and fine-tuned following the framework described in VERL. The model is trained for 30 epochs on 128 H20 GPUs using the AdamW optimizer with a learning rate of $1 \times 10^{-6}$. The models are evaluated with EvalScope Team (2024a) with vllm as the LLM/VLM server.

| Hyperparameter | Value |
|---|---|
| **Data Configuration** | |
| Train Batch Size | 4096 |
| Validation Batch Size | 256 |
| Max Prompt Length | 1024 |
| Max Model Context Length | 2048 |
| **Optimization** | |
| Learning Rate | 1e-6 |
| PPO Mini Batch Size | 2048 |
| Gradient Accumulation | 1 |
| KL Loss Weight | 0.001 |
| **Rollout Configuration** | |
| Rollout Name | vllm |
| Number of Rollouts | 8 |
| **Training & Logging** | |
| Validation Steps | 20 |
| Total Epochs | 30 |

Table 4: Configuration for text-only training.

| Hyperparameter | Value |
|---|---|
| **Data Configuration** | |
| Train Batch Size | 1024 |
| Validation Batch Size | 256 |
| Max Prompt Length | 1024 |
| Max Model Context Length | 2048 |
| **Optimization** | |
| Learning Rate | 1e-6 |
| PPO Mini Batch Size | 1024 |
| Gradient Accumulation | 1 |
| KL Loss Weight | 0.001 |
| **Rollout Configuration** | |
| Rollout Name | vllm |
| Number of Rollouts | 8 |
| **Training & Logging** | |
| Validation Steps | 20 |
| Total Epochs | 30 |

Table 5: Configuration for vision training.

## C  Guidance Prompts

We present the heuristic guidance prompts used in this works.

> **Guidance Prompts**
>
> Prompts we use to trigger model self-reflection.
>
> - Wait, I find a mistake in my previous answer. Let me correct it.
> - Let me double check my previous answer.
> - Hmm, I think I need to revisit my last response.
> - Wait, that doesn't seem quite right. I'll take another look.
> - Hold on, I might have misunderstood the question.
> - Hmm, let me reassess what I just said.
> - Wait a second, there's an error in my logic.
> - Hold up, I think I made a miscalculation there.
> - Hmm, something about that answer feels off. Let me review it.
> - Wait, I just realized I overlooked something important.
> - Hold on, I think there's a better way to explain that.
> - Hmm, now that I think about it, that might not be accurate.
> - Wait, I need to correct myself.
> - Oh, I see a flaw in my previous explanation. Let me fix it.
> - Hold on, I should verify that before continuing.
> - Hmm, I might have jumped to conclusions too quickly.
> - Wait a minute, I need to clarify what I meant earlier.
> - Let me take a moment to re-evaluate my last response.
> - Hold on, I think I need to rephrase that for better accuracy.
> - Hmm, I think I missed an important detail. Let me go back.
>
> This design allows the agent to perform more accurate and interpretable decision-making.

## D  VISUALIZATION

In this section, we attach more visualization of the model reflection ability, see Figure 11. We also demonstrate cases where the reflection mechanism fails to work, see Figure 12.

In Fig. 12, reflection failures commonly arise from two patterns: (1) reflection that starts from an incorrect hypothesis and remains incorrect—confirmation bias leads the model to re-justify the same flawed computation or logic, making only local edits that do not change the final answer; (2) misperception of the evidence due to hallucination, incorrect reading of visual elements like axes and markers, or fabricated details not present in the figure, so subsequent reflection propagates the same faulty observations. A reason for such reflection failure might be due to the reward hack where the model output a correct final answer for verification, but wrong reasoning paths, which usually happen in binary/multiple choices questions.

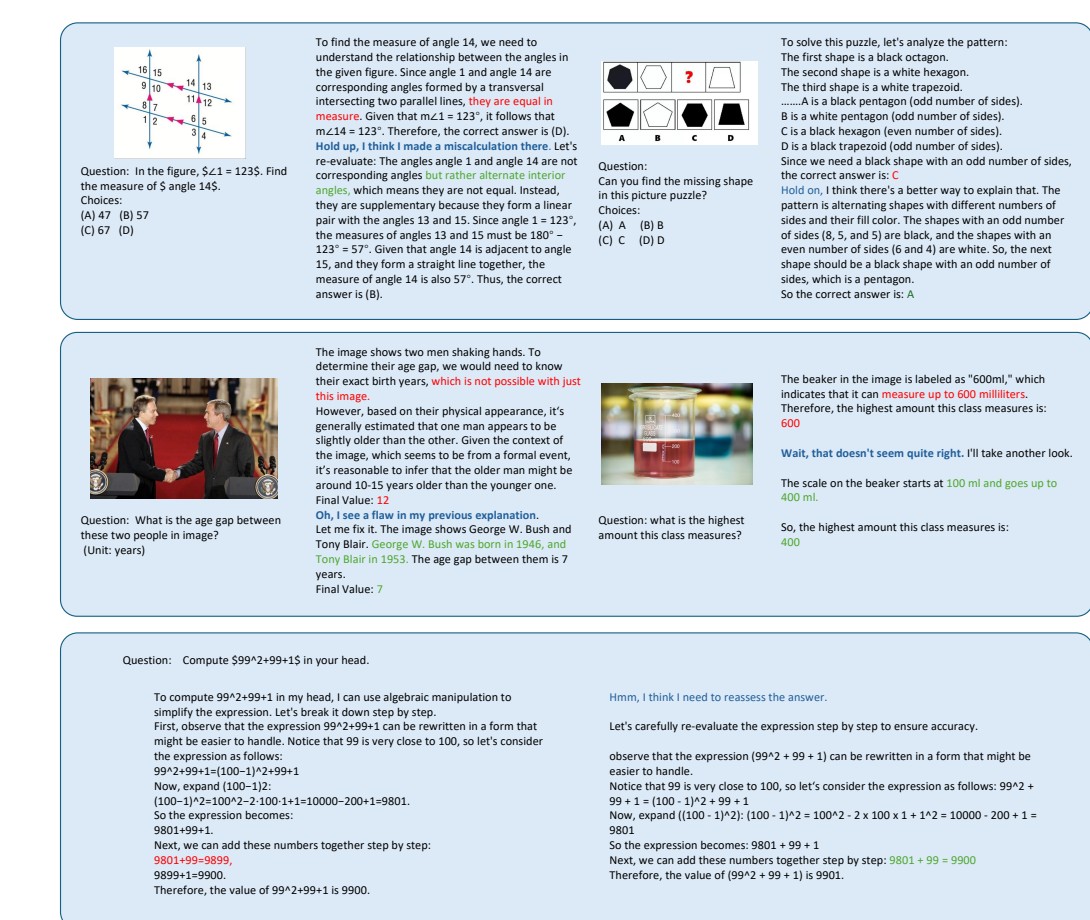

Figure 11: Successful reflection for vision reasoning and text reasoning tasks.

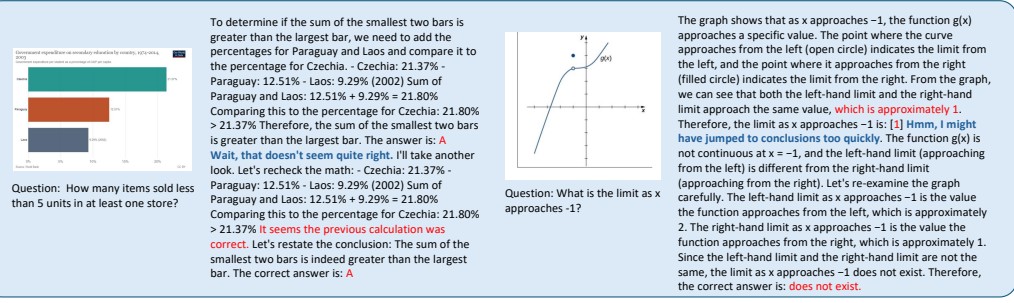

Figure 12: Failed reflection examples.

