# OpenReview forum: "Guided Sampling in Reinforcement Learning for LLM Reasoning"
_ICLR.cc/2026/Conference — Submitted to ICLR 2026_

### Official Review · Reviewer_fwUK · 2025-10-24

**Soundness:** 3
**Presentation:** 3
**Contribution:** 2
**Rating:** 4
**Confidence:** 3

**Summary:**

This paper works on explicitly imparting the reflection ability into policies during RL training. The models are prompted to check their wrong responses and regenerate a new response.

**Strengths:**

+ It is new to explicitly incorporate the reflection process into the RL training for vision-language reasoning tasks.
+ The gains are consistent on multiple benchmarks (e.g., MathVerse and MathVista).
+ The authors provide both qualitative and quantitative analyses suggesting the model learns to reflect and correct prior errors.

**Weaknesses:**

+ at least one model beyond Qwen series should be reported. it is thus unclear whether the success is specific to the Qwen model families.
+ the training time of different methods should be reported, as the proposed sequential method increases the sampling budget. The reflection naturally needs to be conducted after one attempt is finished, and cannot benefit from parallel sampling.
+ The proposed approach benefits from intermediate verifier feedback unavailable to baseline methods, which might partly explain its advantage.
+ (minor) the subfigures in figure 3 are hard to read.

**Questions:**

see Weaknesses

---

### Official Review · Reviewer_vZ34 · 2025-10-28

**Soundness:** 2
**Presentation:** 3
**Contribution:** 2
**Rating:** 4
**Confidence:** 5

**Summary:**

This paper introduces Guided Sampling (GS), a reinforcement learning training framework designed to enhance the self-correction ability of large language models (LLMs) during reasoning. Instead of discarding incorrect responses, GS treats them as intermediate states and appends a reflection prompt (e.g., “Wait, let me re-evaluate…”) to trigger the model to reflect and revise its answer within the same reasoning trace.

Key contributions are as follows:
1. Proposes Guided Sampling, a novel RL sampling pipeline that explicitly trains models to reflect on failures;
2. Designs segment-level rewards to separately credit initial reasoning and subsequent reflection;
3. Demonstrates consistent improvements over GRPO and Best-of-N across both text-only (GSM8K, MATH) and multimodal (MathVista, MathVerse, MM-Math) reasoning benchmarks.

**Strengths:**

S1: This paper addresses a key limitation of current RL methods (e.g., GRPO) that optimize only for final answer correctness, ignoring the reasoning process and self-correction behavior.

S2: This paper formulates reflection as a sequential MDP, where failed attempts become intermediate states, and reflection is triggered by lightweight prompts. This is intuitively reasonable and easy to implement.

S3: This paper conducts extensive experiments on both text-only and multimodal math reasoning tasks, showing consistent gains across 3B and 7B model sizes.

S4: This paper systematically ablates components (guidance, clipping, segment-level advantages), showing incremental benefits from each design choice.

S5: This paper provides detailed analysis of training dynamics, including reward curves, KL divergence, clip magnitudes, and response length, showing that GS encourages longer and more deliberate reasoning.

**Weaknesses:**

W1: While the idea of training reflection is useful, the core mechanism—appending a prompt after failure to encourage revision—is conceptually similar to prior work like Self-Refine, Reflexion, or iterative prompting strategies.

W2: The method assumes access to reliable feedback (e.g., math answer verifiers) at training time. This limits generalizability to open-ended or subjective tasks where such verifiers are unavailable.

W3: Although the paper shows higher reflection success rates, it lacks deep semantic analysis of whether the model truly understands and fixes its mistakes or just surface-level corrections.

W4: Main comparisons are limited to GRPO and Best-of-N. Missing comparisons with stronger self-correction or RL methods (e.g., Reflexion, Self-Refine, RL with self-generated critiques).

W5:  The main baseline is Qwen2.5/Qwen2.5-VL. Besides the Qwen series, if this paper considers other thinking/no-thinking series (such as Deepseek and Llama), will there be a performance improvement similar to that of Qwen.

W6: Only one reflection step is allowed per query. This restricts the potential of the method and avoids harder questions like whether the model can iteratively improve over multiple steps.

**Questions:**

Q1: What happens if the intermediate verifier is wrong (e.g., marks a correct answer as incorrect)? Could this mislead the model into unnecessary or even harmful reflection?

Q2: How would the method perform on open-ended or subjective tasks (e.g., coding, Search)? How would you design reflection triggers without verifiable rewards?

Q3: Could the model learn to intentionally generate wrong answers first and then “correct” them to maximize reward? Are there mechanisms to prevent such reward hacking?

Q4: Why is reflection limited to only one step? Is there a performance saturation or training instability issue with multi-turn reflection?

Q5: How to define a segment, what is the difference between it and splitting step/chunking, and how to determine whether the content of a segment is reasonable?

Q6: Can GS enhance the performance of other thinking/no-thinking models?

---

### Official Review · Reviewer_ET5k · 2025-10-30

**Soundness:** 2
**Presentation:** 3
**Contribution:** 2
**Rating:** 4
**Confidence:** 3

**Summary:**

This paper introduces Guided Sampling (GS), a  RL sampling framework designed to improve self-reflection and error correction capabilities in LLMs. Unlike standard RL approaches that rely solely on outcome-based rewards, GS redefines exploration as a sequential, reflective process, where incorrect model outputs trigger a natural language prompt for reflection and regeneration. The method introduces segment-level credit assignment, separating initial reasoning and reflection, and applies relaxed clipping to stabilize optimization on reflection segments. Experimental results demonstrate consistent and significant improvements over GRPO across both text-only and multi-modal reasoning benchmarks.

**Strengths:**

- The paper addresses a critical limitation in RL for LLMs: the lack of explicit training for self-correction within a single reasoning trajectory.
   - The method significantly improves performance across multiple benchmarks. Gains are especially notable in harder problems.

**Weaknesses:**

- Since reflection involves an additional round of sampling, it inevitably increases computational time. From a training efficiency perspective, is the proposed method truly more efficient than GRPO?
- The experiments are conducted on instruct-tuned models. Would the proposed method still be effective for reasoning-oriented models that already possess strong reflection capabilities?

**Questions:**

- The evaluation is conducted exclusively in the math domain. How well does the method generalize to other domains after training?

---

### Official Review · Reviewer_G9gM · 2025-10-31

**Soundness:** 3
**Presentation:** 3
**Contribution:** 3
**Rating:** 6
**Confidence:** 3

**Summary:**

The paper proposes Guided Sampling (GS), a novel reinforcement learning (RL) framework that enhances reflection-driven reasoning in large language models (LLMs). Traditional RL methods such as PPO and GRPO optimize outcome-based rewards that emphasize correctness of the final answer but fail to explicitly train the model’s ability to self-diagnose and correct reasoning errors mid-trajectory.

The authors reframe exploration as a sequential reflective process: when a model produces an incorrect answer, it receives a “reflection trigger” cue (e.g., “Wait, that doesn’t seem right”) prompting it to reassess and continue reasoning. This transforms the sparse terminal reward structure into dense mid-trajectory credit assignment, explicitly optimizing both “thinking” and “reflection” behaviors.

**Strengths:**

1. Soundness

The authors provide: A clear formulation of multi-turn refinement loops and reward routing. Careful reward normalization and advantage assignment to prevent interference between reasoning and reflection tokens. These improvements are well-motivated and grounded in prior RLVR and GRPO analyses.

2. Strong Empirical Validation

Results are substantial and well-presented: On MathVista/MathVerse/MM-Math, GS yields +2–3 pp average gains over GRPO. Enough ablation is done regarding the training details in Section 3.3

**Weaknesses:**

1. Ablation Scope

It lakes ablation on other alternative training details, such as other ways to do reward normalization (e.g. per segment) together with segment/trajectory-level advantage assignment.

**Questions:**

None

---

### Meta-Review · Area_Chair_QYrW · 2025-12-21

**Summary:**

The common issues lied in:
1. Similar idea appears in prior work like Self-Refine, Reflexion, or iterative prompting strategies, and the paper failed to clarify the contribution.
2. Poor generalization performance and computation burden might be raised since it introduced extra reflection in the training.
3. It lacked extensive convinced experiments compared to more state-of-the-art methods on broader benchmarks.

The authors did not reply to the review comments.

It has to be rejected.

**Reviewer Concerns:**

No rebuttals.

**Reviewer Scores:**

It received 4 reviews with the average score 4.5.

No discussions for no rebuttals.

---

### Decision · Program_Chairs · 2026-01-26

Reject